# PHYSICS-INFUSED INTENTION NETWORK FOR CROWD SIMULATION

## ABSTRACT

Crowd simulation has garnered significant attention in domains including traffic management, urban planning, and emergency management. Existing methods can be classified as either rule-based or learning-based approaches, with the former lacking authenticity and the latter lacking generalization. Recent research has attempted to combine these approaches and propose physics-infused methods to address the aforementioned limitations. However, they continue to adhere strictly to the framework of the physical model, neglecting to depict the attention mechanism as a critical component of behavior. This limitation results in deficiencies in both the fidelity and generalizability of the simulations. This paper introduces a novel framework called Physics-infused Intention NEtwork (PINE) for crowd simulation. Our model introduces a physical bias while endowing pedestrians with the ability to selectively enhance the fine-grained information most relevant to one's current behavior. In addition, we design a variable-step rollout training approach with an optimized loss function to address cumulative errors in simulation. By conducting extensive experiments on four publicly available real-world datasets, we demonstrate that our PINE outperforms state-of-the-art simulation methods in accuracy, physical fidelity, and generalizability.

## 1 INTRODUCTION

Crowd simulation is a process of simulating the movements of a large number of individuals in specific scenarios, with a focus on interaction dynamics. It serves as the foundation of various important applications, including facility, infrastructure, and building design (Bitgood, 2006), crowd evacuation (Boltes et al., 2018), autonomous driving cars (Poibrenski et al., 2020), human-robot interactions (Tai et al., 2018), assistive technologies in industrial scenarios (Leo et al., 2017), and entertainment (e.g., augmented and virtual reality) (Rodin et al., 2021).

Current crowd simulation methods can be broadly categorized into two main types: rule-based and learning-based. Rule-based methods use a set of expert knowledge-based rules to simulate crowd dynamics. These methods exhibit robust performance across various scenarios but overlook the heterogeneity of pedestrian behavior. Previous studies (Cao et al., 2016; Subaih et al., 2020) have shown that pedestrian behavior is influenced by multiple factors such as age, gender, and culture. Uniform physical rules fail to effectively capture these individual differences. Additionally, limited rules struggle to accurately represent the complex interactions and motion patterns of crowds in real-world scenarios. On the other hand, learning-based methods are able to model more intricate behavioral patterns when provided with ample data. However, it is important to note that these methods often face challenges when it comes to generalizing to long-term simulations or different scenarios. This limitation has been highlighted in related studies (Zhang et al., 2022). Considering the complementarity of rule-based and learning-based methods, recent researchers have started to explore the integration of these two approaches by proposing physics-informed deep learning frameworks. They replace key components of the rule-based model, such as partial or complete set of parameters of force components in SFM (Social Force Model), with neural networks and train on real data. It is important to note that they adhere to the characterization framework of pedestrian behavior patterns in the physical model, wherein pedestrians assign equal importance to all factors. Nevertheless, the human brain operates as a limited information system. Humans possess behavioral intention that selectively prioritizes information that aligns with their present behavioral objectives (Buschman & Kastner, 2015).

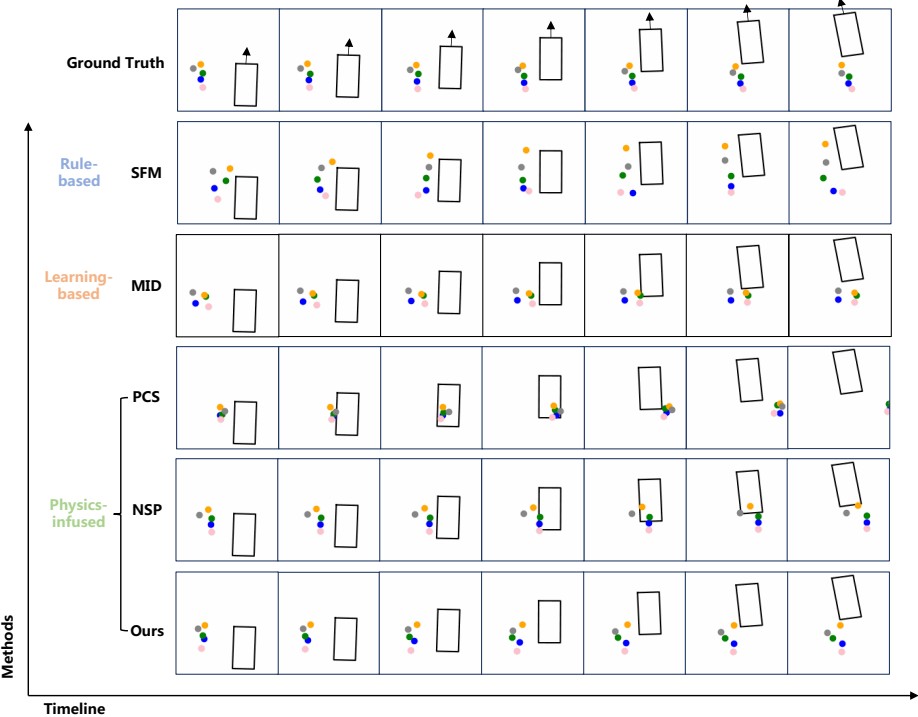

Figure 1: A visualization of Human-Car interaction scenario. Different colored dots represent different pedestrians, while the rectangular box represents vehicles. The black arrow indicates the forward direction of the vehicle. The horizontal axis represents the time steps, increasing from left to right. The vertical axis represents the simulated effects presented by different methods in this scenario.

In order to incorporate the impact of behavioral intention, we propose a novel framework that enables pedestrians to adaptively adjust the influence of different factors on their behavior. This approach facilitates the achievement of accurate and physically consistent simulation outcomes. Fig. 1 illustrates the simulation effects of various methods in a person-vehicle interaction scenario. In the rule-based simulation (SFM (Helbing & Molnar, 1995)), pedestrians consistently allocate equal attention to the attraction of distant goals and the repulsion exerted by nearby vehicles. Consequently, pedestrians repeatedly collide with vehicles, resembling the behavior of a pinball. Learning-based (MID (Gu et al., 2022)) and existing physics-infused (PCS and NSP (Karniadakis et al., 2021; Yue et al., 2022)) methods disregard the influence of vehicles and solely concentrate on reaching the goal. Consequently, pedestrians directly cross through vehicles. In contrast, our proposed method incorporates an attention mechanism that enables pedestrians to dynamically adjust their attention. As pedestrians approach vehicles, they allocate more attention to the vehicles in front of them. Once the vehicles move away, pedestrians redirect their attention towards their intended goals. Through our attention mechanism, pedestrians demonstrate behavior patterns that closely resemble real-life scenarios, encompassing deceleration, avoidance, and resumption of movement.

In this paper, we present a crowd simulation model called Physics-infused Intention NEtwork (PINE). Our model combines the advantages of rule-based and learning-based approaches. By integrating rule-based models to incorporate physical biases and utilizing neural networks to learn complex attention mechanisms of pedestrians, our model provides an accurate, physical consistent and generalized simulation. The contributions of our work are outlined as follows:

1) We propose PINE for crowd simulation. At the input end of the model, we design an adaptive feature extraction module that selectively extracts important features based on the physical model's characterization of the state. At the output end of the model, we introduce an attention module that operates on the raw output of the rule-based model. Additionally, the neural network generates a

residual behavior representing patterns beyond the scope of the rule-based model. The optimized rule model output and neural network output jointly control pedestrian behavior

2) We introduce a Variable-step Rollout Training algorithm to address cumulative errors during simulation. Similiar to (Zhang et al., 2022), we adopt a multi-step rollout training approach in the training phase. Building upon this, taking into account the computational burden and shift phenomenon, we design an automated adjustment of rollout steps and a loss function with a shift regression bias.

3) Extensive experiments demonstrate the accuracy, fidelity, and generalizability of our method in various scenarios.

## 2 RELATED WORK

### 2.1 RULE-BASED CROWD SIMULATION

Rule-based methods typically use calibrated rules to model the interaction between individuals (Helbing & Molnar, 1995; Dietrich & Köster, 2014; Xi et al., 2011) or groups of pedestrians (Hughes, 2002; Karamouzas & Overmars, 2011). One of the most known techniques in this category is the social forces method (Helbing & Molnar, 1995) which models changes in pedestrian behavior based on forces calculated by combining three components : acceleration force which captures an individual desire to reach a certain velocity, repulsive force which shows the tendency of pedestrians to keep a certain distance from their neighboring individuals, and repulsive force from obstacles. This type of approach exhibits good generalization ability, but lacks the capability to represent complex and heterogeneous behaviors of pedestrians.

### 2.2 LEARNING-BASED CROWD SIMULATION

With the emergence of machine learning techniques and increase in computation power, many recent approaches rely on learning-based approaches to learn and predict different patterns of pedestrian behavior (Song et al., 2018; Kouskoulis et al., 2018). For example, (Alahi et al., 2016) use a long short-term memory (LSTM) network with a social pooling layer that extracts interactions among nearby pedestrians, and (Gupta et al., 2018) use Generative Adversarial Networks (GANs) to predict social acceptable trajectories. Other approaches (Sun et al., 2021; Mangalam et al., 2021; Gu et al., 2022) propose other strategies such as diffusion models. However, they still have shortcomings in terms of generalization ability.

### 2.3 PHYSICS-INFUSED CROWD SIMULATION

Due to the complementarity of rule-based and learning-based methods, recent researchers have made attempts to integrate physical models with neural networks. These attempts involve replacing certain key components of the physical model with neural networks and training them using real data. For example, PCS (Physics-informed Crowd Simulator) (Zhang et al., 2022) replaces the core terms of SFM with GNNs (Graph Neural Networks), and the NSP (Neural Social Physics) model (Yue et al., 2022) designs a SFM with learnable parameters. However, they continue to adhere to the framework of SFM and neglect the attention mechanism of humans, resulting in deficiencies in both fidelity and generalizability.

## 3 PROBLEM FORMULATION

### 3.1 PEDESTRIAN SIMULATION

Given the initial state, the pedestrian simulation aims to simulate the pedestrian behavior of an arbitrary duration. We formulate the simulation as an iterative prediction problem. At each time step $t$, we take pedestrian internal states, including their positions, velocities, accelerations, and destinations, together with the environment external states, including the map information, the obstacles' positions, velocities, and accelerations, as input. The output is the actions that pedestrians need to adopt at the current time frame. After taking actions, we can obtain new states at $t + 1$ from the

environment. Formally, it can be formulated as follows,

$$s_p^{t+1}, s_e^{t+1} = P(s_p^t, s_e^t, f_\theta(s_p^t, s_e^t)) \tag{1}$$

where $s_p^t$ and $s_e^t$ are the the pedestrians' states and the environment states at time $t$. $f_\theta$ outputs the actions of pedestrian and $P$ reflects the dynamics of the environment. To better portray the continuum of pedestrian behavior and generate physically consistent results, $f_\theta$ directly outputs the accelerations of pedestrians. Then the update of crowd can be formulated as follows,

$$p_{t+1} = p_t + v_{t+1}\Delta t, v_{t+1} = v_t + a_{t+1}\Delta t \tag{2}$$

where $p$, $v$ and $a$ represent the position, velocity and acceleration of the pedestrian, respectively.

## 3.2 PHYSICAL MODEL

The Social Force Model (SFM) was established by Helbing et al. according to the Newton's second law of motion (Helbing & Molnar, 1995). The model assumes that pedestrians are influenced by three factors: mentality, other pedestrians and environment. The influences of the three factors can be respectively measured by three forces: the self-driving force, the acting force between two pedestrians and the acting force between a pedestrian and obstacles, which can be formulated as follows,

$$m_i \boldsymbol{a}_i = \boldsymbol{f}_{iD} + \sum_{j \neq i, j \in \mathcal{P}} \boldsymbol{f}_{ji} + \sum_{o \in \mathcal{O}} \boldsymbol{f}_{oi} \tag{3}$$

$$\boldsymbol{f}_{iD} = m_i \frac{v_{id}\boldsymbol{n}_{iD} - \boldsymbol{v}_i}{\tau} \tag{3a}$$

$$\boldsymbol{f}_{ji} = \lambda_1 e^{-d_{ji}/\alpha_1} \cdot \boldsymbol{n}_{ji} \tag{3b}$$

$$\boldsymbol{f}_{oi} = \lambda_2 e^{-d_{oi}/\alpha_2} \cdot \boldsymbol{n}_{oi} \tag{3c}$$

where $\mathcal{P}$ and $\mathcal{O}$ refer to the set of pedestrians and obstacles, respectively. $\boldsymbol{f}_{iD}$, $\boldsymbol{f}_{ji}$ and $\boldsymbol{f}_{oi}$ represent the traction force of destination $D$, the repulsive force of pedestrian $j$ and obstacle $o$ on pedestrian $i$, respectively. The magnitude and direction of the traction force $\boldsymbol{f}_{iD}$ depend on the desired walking velocity $v_{id}\boldsymbol{n}_{iD}$, where $v_{id}$ denotes the speed and $\boldsymbol{n}_{iD}$ denotes the unit vector to the target direction, and the current velocity $\boldsymbol{v}_i$ of pedestrian $i$. $\tau$ is the simulation time step. The repulsive force between pedestrian $i$ and other pedestrians or obstacles is correlated with the relative distance between them, denotes as $d_{ji}$ and $d_{oi}$, and in the direction of their relative position, denotes as $\boldsymbol{n}_{ji}$ and $\boldsymbol{n}_{oi}$. $\lambda_1$, $\lambda_2$, $\alpha_1$ and $\alpha_2$ are tunable parameters with a physical meaning. Since these parameters can greatly affect the effectiveness of SFM, we adopt the optimized form (Moussaïd et al., 2009), where the parameters have been optimized in controlled experiments.

# 4 METHOD

## 4.1 FRAMEWORK

In this section, we present PINE, which simulates crowd behaviour based on the physical model and neural networks. As shown in Fig. 2 (a), at each time step $t$, we construct a field of view for each pedestrian with their current position as the center, a certain radius, and angle. Only pedestrians or obstacles appearing within the field of view will be considered as features and constitute the current state of pedestrians. We propose two modules, namely the Feature Adaptive Extraction Module (FAEM) and the Force Intention Module (FIM), which introduce physical biases and attention mechanisms. Moreover, we generate an additional Residual Action (RA) at the output to compensate for behavior patterns that cannot be accurately captured by the physical model. To address the cumulative error that occurs during simulation, we introduce a Variable-step Rollout Training Algorithm (VRTA).

## 4.2 FEATURE ADAPTIVE EXTRACTION MODULE (FAEM)

The raw information received by the neural network may contain a lot of noises and irrelevant contents. Inputting all contents indiscriminately into the neural network can hinder its learning process.

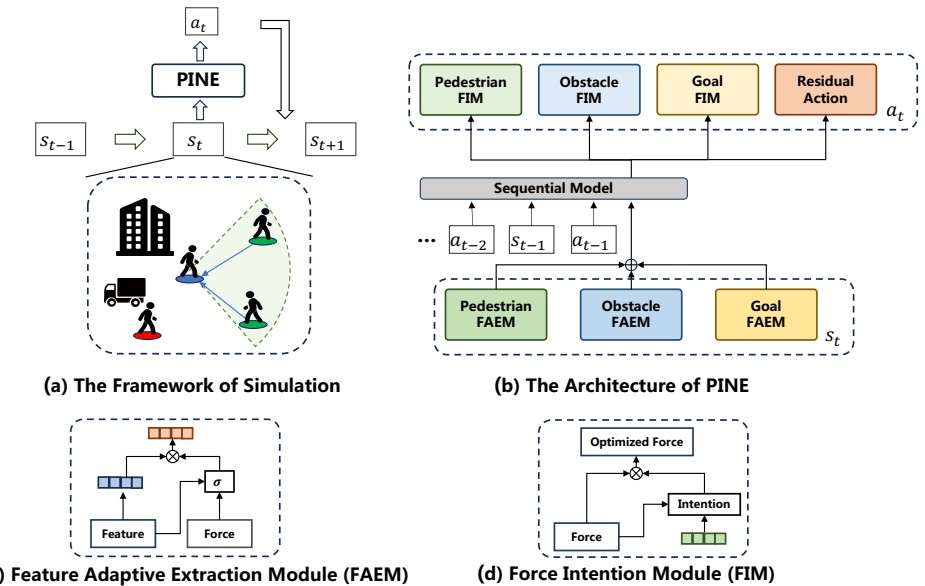

(a) The Framework of Simulation    (b) The Architecture of PINE

(c) Feature Adaptive Extraction Module (FAEM)    (d) Force Intention Module (FIM)

Figure 2: The simulation framework and the architecture of our proposed PINE are described. PINE is designed to receive information about the current visual field of pedestrians and generate corresponding actions. The model utilizes an abstract representation of the environment through a physical model, which helps extract critical information from raw data. The spatio-temporal information collected by the sequential model is then used to direct pedestrians' attention to factors that require more focus in the current context. Additionally, the neural network generates a residual action to cooperatively control pedestrian behavior.

The intermediate physical quantities of the physical model, on the other hand, can more accurately reflect the influence of current factors on pedestrian behavior. Specifically, in Social Force Model (SFM) (Helbing & Molnar, 1995), the magnitude and direction of the forces directly reflect the influence of corresponding factors on pedestrians. For example, when the repulsive force towards obstacles in SFM is high, the neural network should receive more raw information related to obstacles. Therefore, as shown in Fig. 2(c), we construct gate units for corresponding features based on the SFM forces at the input end of the features as follow:

$$z_t = i_t^p x_t^p + i_t^o x_t^o + i_t^g x_t^g \tag{4}$$

$$i_t^p = \sigma(W^p \cdot [x_t^p, f_t^p] + b^p) \tag{4a}$$

$$i_t^o = \sigma(W^o \cdot [x_t^o, f_t^o] + b^o) \tag{4b}$$

$$i_t^g = \sigma(W^g \cdot [x_t^g, f_t^g] + b^g) \tag{4c}$$

where $x_t^p$, $x_t^o$ and $x_t^g$ respectively denote the raw feature of the surrounding pedestrians, obstacles and goals at time $t$. $f_t^p$, $f_t^o$ and $f_t^g$ respectively denote the force components of the surrounding pedestrians, obstacles and goals at time $t$. $W$ and $b$ are learnable parameters and $\sigma$ is the sigmod function. $i^p$, $i^o$ and $i^g$ are used to adjusting the weights of different input features, respectively. $z_t$ represents the final input features to the sequential model at time $t$. These gate units adjust the input magnitude of different features, helping the neural network learn the corresponding behavior on features it needs to focus on.

### 4.3 FORCE INTENTION MODULE (FIM)

The physical model possesses stronger generalization capability and maintains physical consistency. Therefore, we preserve the output of the SFM at the output end to inherit the excellent properties of the physical model. However, the original output of the physical model has flaws in terms of the realism in simulations. For example, SFM fails to capture the behavior of pedestrians gradually slowing down or stopping while waiting for vehicles to pass. Instead, it continues to exert force,

resulting in pedestrians continuously colliding with vehicles. The main reason is that in SFM, different factors act independently on pedestrians. When pedestrians are obstructed by vehicles, they still maintain the drive to reach their destination. However, pedestrians possess attention mechanisms. They have behavioral intention which selectively prioritizes information that is most relevant to their current behavioral goals. In order to incorporate attention mechanisms into SFM, we design the Force Intention Module (FIM) to adjust the force components in SFM as follow:

$$f_t = e_t^p f_t^p + e_t^o f_t^o + e_t^g f_t^g \tag{5}$$

$$e_t^p = \sigma(W^p \cdot [h_t, f_t^p] + b^p) \tag{5a}$$

$$e_t^o = \sigma(W^o \cdot [h_t, f_t^o] + b^o) \tag{5b}$$

$$e_t^g = \sigma(W^g \cdot [h_t, f_t^g] + b^g) \tag{5c}$$

where $h_t$ represents the output of the sequential model at time $t$, $e_t^p$, $e_t^o$ and $e_t^g$ denote the attention on force components $f_t^p$, $f_t^o$ and $f_t^g$, respectively. $f_t$ represents the optimized force after the FIM.

## 4.4 RESIDUAL ACTION (RA)

SFM only considers the influence of three limited factors on pedestrian behavior. However, pedestrian behavior is complex and influenced by a multitude of underlying factors, such as pedestrians' psychological state. In order to compensate for the shortcomings of SFM in characterizing the impact of these latent features on pedestrians, our model directly outputs an additional Residual Action (RA) from the sequential model based on the information of historical trajectories, which is computed as follow

$$a_t^{RA} = MLP(h_t) \tag{6}$$

where $h_t$ represents the output of the sequential model at time $t$ and $a_t^{RA}$ represents the RA at time $t$. In contrast to FIM, where all forces are determined by known factors, RA can capture the the influence of latent features on pedestrian. FIM and RA collectively govern pedestrian behavior.

## 4.5 VARIABLE-STEP ROLLOUT TRAINING ALGORITHM (VRTA)

Employing single-step training strategy on a limited set of trajectory data would not be sufficient for conducting long trajectory simulations. A recognized problem is the compounding errors that occur when deviating from the trajectory samples during simulation. An effective solution is to employ rollout training (Ross & Bagnell, 2010), wherein the model simulates $T$ steps and utilizes the results collectively to compute losses and update model parameters. Similar strategies have also been implemented in previous studies (Zhang et al., 2022). Considering the inefficiency of rollout training, it is usually to combine single-step and multi-step approaches in practical applications. However, the existing training strategy still has two significant limitations: 1) it initiates iteration at a fixed length, unable to achieve a balance between model performance and computational efficiency, and 2) it neglects the regression of the shift. To overcome the aforementioned limitations, we propose the Variable-step Rollout Training Algorithm (VRTA). Firstly, we incrementally increase the value of $T$, allowing the model to improve simulation results on shorter time steps before extending the step length. This approach enhances training efficiency. Secondly, we optimized the loss function to minimize the model's shift, which is computed as follows:

$$\mathcal{L}_F = \sum_{t=1}^{T} \lambda^t ||f_\theta(\hat{s}_t, \tau_t) - a_t - \sum_{i=1}^{t} i \cdot (a_{t-i} - \hat{a}_{t-i})||^2 \tag{7}$$

$$\tau_t = (\hat{a}_{t-1}, \hat{s}_{t-1}, \hat{a}_{t-2}, \hat{s}_{t-2}, ..., \hat{a}_1, s_1) \tag{8}$$

where $\tau_t$ represents the simulation trajectory before time $t$ and $\hat{s}_t$ represents the current state at time $t$. $f_\theta(\hat{s}_t, \tau_t)$ generates the predicted action $\hat{a}_t$ and $a_t$ denotes to the true action at state $s_t$. Previous work solely focused on the loss function between $f_\theta(\hat{s}_t, \tau_t)$ and $a_t$, disregarding the differences between $\hat{s}_t$ and $s_t$. In other words, pedestrian cannot transition to $s_{t+1}$ by performing $a_t$ at $\hat{s}_t$. Therefore, we introduce the additional bias $\sum_{i=1}^{t} i \cdot (a_{t-i} - \hat{a}_{t-i})$. The additional bias represents the accumulated error between predicted actions and true actions up to the current time step $t$, which helps the pedestrian transition back to $s_{t+1}$ at the next step. To prevent excessive behavioral distortions, we introduce a weighting factor $0 < \lambda < 1$ that emphasizes actions made in states closer to the initial steps, which are closer to the ground truth. The derivation process of the loss function can be referred to in the appendix.

| Groups | Models | DUT | | | | ETH | | | |
|---|---|---|---|---|---|---|---|---|---|
| | | MAE | DTW | OT | MMD | MAE | DTW | OT | MMD |
| Rule-based | SFM | 1.509 | 0.506 | 2.865 | 0.033 | 0.816 | 0.334 | 1.262 | 0.200 |
| Learning-based | PCCSNet | 8.009 | 7.978 | 32.98 | 0.588 | 3.799 | 3.624 | 10.97 | 0.907 |
| | Y-net | 2.658 | 2.122 | 8.173 | 0.096 | 1.682 | 1.415 | 4.148 | 0.29 |
| | MID | 2.653 | 2.300 | 7.711 | 0.108 | 2.246 | 2.008 | 7.654 | 0.724 |
| Physics-infused | PCS | 1.851 | 0.894 | 3.249 | 0.062 | 1.265 | 0.738 | 1.997 | 0.303 |
| | NSP | 0.860 | 0.455 | 1.702 | 0.018 | 0.590 | 0.350 | 0.986 | 0.147 |
| | Ours | **0.781** | **0.396** | **1.433** | **0.014** | **0.430** | **0.308** | **0.448** | **0.109** |
| | $\Delta$ | (+9.2%) | (+13%) | (+16%) | (+22%) | (+27%) | (+12%) | (+55%) | (+26%) |

Table 1: The performance evaluation results on the DUT and ETH datasets. The training and testing of the model are conducted on the same dataset.

## 5 EXPERIMENTS

### 5.1 EXPERIMENT SETUP

**Dataset**. We conduct crowd simulation evaluation experiments of the model on four open-source datasets: the DUT (Yang et al., 2019) dataset, the ETH (Pellegrini et al., 2009) dataset, the GC (Zhou et al., 2011) dataset and the UCY (Lerner et al., 2007) dataset. The four datasets differ in scenarios, scale, duration, and pedestrian density, allowing us to verify the generalization performance of the model.

**Baseline Methods**. We compare our framework with six state-of-the-art models from three categories, including rule-based models, learning-based models and physical-infused models. Rule-based model adopts the Social Force Model (SFM) (Moussaïd et al., 2009), which is widely used in the state-of-the-art commercial crowd simulators. Learning-based methods include PCCSNet (Sun et al., 2021), Y-net (Mangalam et al., 2021) and MID (Gu et al., 2022), which are the state-of-the-art methods based on different framework to predict pedestrian trajectories. Physical-infused models include PCS (Zhang et al., 2022) and NSP (Yue et al., 2022), which both combine physical models with neural networks to enhance pedestrian simulation effectiveness.

**Experiment Settings and Reproducibility**. For the DUT and ETH datasets, we divided the dataset into training, valid and testing sets in a 4:1:1 ratio based on the time scale to validate the effectiveness of our method in training and simulating on similar datasets. For the GC and UCY datasets, we used them as test sets to validate the transferability of our method in simulating pedestrian behavior after training on DUT and ETH datasets. We use four widely adopted metrics, including the Mean Absolute Error (MAE) and Dynamic Time Warping (DTW) characterizing microscopic simulation accuracy, Optimal Transport divergence (OT) and Maximum Mean Discrepancy (MMD), measuring macro-level distribution differences. We perform a grid search on all the hyperparameters for all models. For reproducibility, we make our codes available.

### 5.2 OVERALL PERFORMANCE COMPARISON

To examine the effectiveness of our proposed framework, we compare the performance of the PINE with different types of state-of-the-art baselines on two different scenario large-scale datasets in Table 5.2. Here, we summarize key observations and insights as follows:

**The Superior Performance of PINE**: PINE outperforms all state-of-the-art baselines across all four evaluation metrics on both datasets. Specifically, it provides a relative performance gain of 9.2%, 13%, 16%, and 22% on the DUT dataset and 27%, 12%, 55%, and 26% on the ETH dataset, in terms of MAE, DTW, OT, and MMD, respectively, which demonstrates the effectiveness of our proposed model. Our model gives accurate simulations from both micro trajectory and macro distribution perspectives.

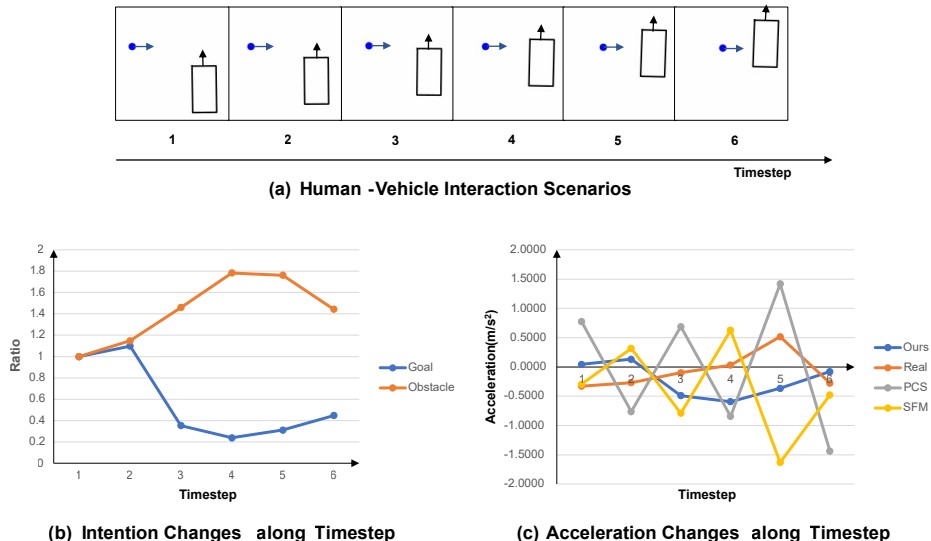

Figure 3: Scenarios involving interactions between individual pedestrian and vehicle. As pedestrians approach vehicles, their attention towards obstacles increases while their attention towards reaching the goal decreases. Our model is able to output fidelity changes in acceleration.

| Groups | Models | GC | | | | UCY | | | |
|---|---|---|---|---|---|---|---|---|---|
| | | MAE | DTW | OT | MMD | MAE | DTW | OT | MMD |
| Rule-based | SFM | 1.996 | 0.940 | 5.939 | 0.209 | 1.487 | 0.751 | 4.086 | 0.711 |
| Learning-based | PCCSNet | 9.591 | 9.509 | 44.77 | 0.810 | 8.298 | 8.296 | 94.56 | 2.288 |
| | Y-net | 8.594 | 7.842 | 68.80 | 0.704 | 2.535 | 2.147 | 11.56 | 0.955 |
| | MID | 5.335 | 5.108 | 17.79 | 0.407 | 2.435 | 2.243 | 8.503 | 1.025 |
| Physics-infused | PCS | 2.357 | 1.286 | 4.891 | 0.188 | 1.481 | 0.754 | 3.669 | 0.796 |
| | NSP | 2.501 | 1.459 | 6.346 | 0.203 | 1.039 | 0.737 | 2.302 | 0.529 |
| | Ours | **1.309** | **0.832** | **3.240** | **0.139** | **0.807** | **0.492** | **1.839** | **0.393** |
| | Δ | (+34%) | (+11%) | (+43%) | (+32%) | (+26%) | (+29%) | (+20%) | (+19%) |

Table 2: The generalization performance results on the GC and UCY datasets. The training is conducted on DUT and ETH datasets.

**Intention Analysis and Its Effect**: To further explore how pedestrian intentions change in response to different scenarios, we conducted a case study using our model's rollout simulations involving human-vehicle interactions shown in Fig. 3 (a). We plot the changes of intention corresponding to goal and obstacle respectively in Fig. 3 (b). The calculation method for intention involves dividing the current time neural network output weights by the output weights at timestep 1. As we can see, as the pedestrian approaches vehicles, their intention towards goal weakens, while their intention towards obstacle avoidance strengthens. This kind of change aligns with the realistic pedestrian behavior, where they pay more attention to higher priority matters. As shown in Fig 3 (c), Intention mechanism helps our model generate smoother acceleration changes, which aligned with real-world situations. On the other hand, SFM and PCS which lack dynamic driving exhibit noticeable oscillations in acceleration changes that do not align with real-world situations.

## 5.3 GENERALIZABILITY

To examine whether our proposed method can generalize well beyond its training distributions, we test the performance of all methods trained on the previous two datasets on GC and UCY datesets.

| Models | DUT | | | | ETH | | | |
|---|---|---|---|---|---|---|---|---|
| | MAE | DTW | OT | MMD | MAE | DTW | OT | MMD |
| Ours | **0.781** | **0.396** | **1.433** | **0.014** | **0.446** | **0.313** | **0.465** | 0.122 |
| w/o FAEM | 1.096 | 0.577 | 1.971 | 0.028 | 0.553 | 0.429 | 0.601 | 0.132 |
| w/o FIM | 1.100 | 0.764 | 2.413 | 0.018 | 0.747 | 0.664 | 1.233 | 0.185 |
| w/o RA | 1.040 | 0.630 | 1.943 | 0.026 | 0.643 | 0.522 | 0.912 | **0.117** |
| w/o VSRT | 0.974 | 0.581 | 1.979 | 0.023 | 0.547 | 0.441 | 0.881 | 0.147 |

Table 3: The ablation performance results on the DUT and ETH datasets. The training and testing of the model are conducted on the same dataset.

| Models | GC | | | | UCY | | | |
|---|---|---|---|---|---|---|---|---|
| | MAE | DTW | OT | MMD | MAE | DTW | OT | MMD |
| Ours | **1.309** | 0.832 | **3.240** | **0.139** | **0.807** | **0.492** | **1.839** | **0.393** |
| w/o FAEM | 1.629 | 1.089 | 3.675 | 0.144 | 0.975 | 0.703 | 2.42 | 0.504 |
| w/o FIM | 1.944 | 1.448 | 8.178 | 0.215 | 1.207 | 0.964 | 3.206 | 0.501 |
| w/o RA | 1.455 | **0.829** | 3.378 | 0.139 | 0.995 | 0.567 | 2.323 | 0.503 |
| w/o VSRT | 1.565 | 1.079 | 4.397 | 0.159 | 0.991 | 0.765 | 2.600 | 0.408 |

Table 4: The ablation performance results on the GC and UCY datasets. The training is conducted on DUT and ETH datasets.

Both datasets have a data distribution fundamentally different from the training dataset. We show the results in Table 5.2. PINE outperforms all the baselines on both datasets, which demonstrates their strong generalizability. In particular, it provides a relative performance gain of 34%, 11%, 43%, and 32% on the GC dataset and 26%, 29%, 20%, and 19% on the UCY dataset, in terms of MAE, DTW, OT, and MMD, respectively.

## 5.4 ABLATION STUDY

In order to examine how different parts of our designs, including the Feature Adaptive Extraction Module (FAEM), Force Intention Module (FIM), Residual Action (RA), and Variable-step Rollout Training (VSRT) contribute to the performance, we consider four variants of our proposed methods, including PINE w/o FAEM, PINE w/o FIM, PINE w/o RA, and PINE w/o VSRT. For each of the variants, we remove one of the key designs.

We present the performance results of the aforementioned version in Table 5.4 and generalization performance in Table 5.4. We have two key observations: First, removing any of the components results in a certain level of performance decrease compared with the full model, which suggests that all the designed components are effective. Second, the removal of the FIM, which is key component of attention mechanism, generally leads to the most significant decline in the model's performance. The introduction of physical biases and intention mechanism significantly improve the performance of our model. This further validates the effectiveness of our framework.

## 6 CONCLUSION

This paper presents a novel approach called Physics-infused Intention NEtwork (PINE) for crowd simulation. The approach employs a rule-based model to incorporate physics biases and neural networks to learn the attention mechanism of pedestrians directly from real data. Extensive experiments were conducted to demonstrate the outstanding performance of the proposed model. Currently, the model has been exclusively applied to pedestrian simulation. Furthermore, the proposed framework can be adapted to learn human behavior in various scenarios. Future studies should explore additional scenes to expand the variety of behaviors that the model can simulate.

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

## A APPENDIX

### A.1 DATASET STATISTICS

To evaluate our models, we introduce two large-scale real-world datasets that differ in the scene, scale, pedestrian density, pedestrian demographics, and pedestrian behavior patterns. The basic statistics of all four datasets are shown in Table 5.4, and the details of the datasets are as follows:

Table 5: The snapshots of four datasets.

| Statistics | DUT | ETH | GC | UCY |
|---|---|---|---|---|
| Average duration of a trajectory (s) | 9.967 | 7.859 | 12.859 | 13.178 |
| Average speed ($m \cdot s^{-1}$) | 1.317 | 1.18 | 1.016 | 1.071 |
| Average acceleration ($m \cdot s^{-1}$) | 0.42 | 0.607 | 0.412 | 0.143 |
| Pedestrian density ($m^{-2}$) | 0.073 | 0.054 | 0.028 | 0.032 |

**DUT**: The DUT dataset was collected at two crowded locations in the campus of Dalian University of Technology (DUT) in China. When a crowd of pedestrians interact with a vehicle, there is no priority (the right of way) for either pedestrians or the vehicle. It annotates 1793 pedestrians and moving vehicles. This dataset provides the physical location of pedestrians in real space.

**GC**: The GC dataset is built on a half an hour crowd surveillance video. It annotates the walking routes of 12684 pedestrians on a 30m × 35m square in image coordinates. As our models are based on the physical laws, we need the pedestrian positions in real-world coordinates but not image coordinates. Therefore, we perform a projective transformation to transform the coordinates. The homography matrix is calculated based on the the proposed spatial information in (Schlichting, 2003).

**ETH**: The ETH dataset contains two scens coined as ETH and Hotel. This dataset is extensively used in Human Trajectory Prediction literature. This dataset provides the physical location of pedestrians in real space.

**UCY**: The UCY dataset is composed of three sequences, but as we focus on long-term and complex behavior simulation in this work, we only choose the sequence of Zara. In this scenario, not only are there a large number of long trajectories, but there are also complex behaviors such as sharp turns. These trajectories can better test the performance of the models.

### A.2 PINE IMPLEMENTATION DETAILS

We construct each pedestrian's visual field and employ two different structures to process the raw information within the field of view. For features such as vehicles and pedestrians that can be represented with a single point, we utilize graph structures for processing. For semantic information in the scene, such as walkable and non-walkable areas, we process it using image-based representations. As shown in Fig. 4, the pedestrian's field of view is controlled by parameters $r_p$, $\omega$ , and $r_{env}$. In our experimental setup,, we set the sector-based visual fields' $r_p$ as 4 meters, the angle $\omega$ as 90 degrees and $r_{env}$ as 5 meters.

Our framework supports the integration of multiple sequence models, such as LSTM and Transformer. In our experimental setup, we use Transformer as our sequence model, and the input and output of Transformer are illustrated in Fig. 5. At each time step $t$, we input the current state of pedestrians and their historical trajectory information into Transformer, and output the corresponding latent variables $h_t$ for further processing by FIM and RA.

The training of the model can be divided into two steps, namely Single-step Pre-train and Variable-step Rollout Training Algorithm (VRTA). The differences between the Single-step Pre-train and Rollout Training are illustrated in Fig. 6. Rollout training is closer to real-world simulation scenarios, as the model's input is influenced by the errors generated from its past predictions. The longer the training step, the closer it approximates the real-world simulation scenario. Correspondingly, this training mode also requires more time as it involves continuous simulation to generate new data. In our experiment, we adopt a method of continuously increasing the rollout step to obtain a balance between model performance and computational efficiency. Specifically, we first update the model parameters by performing Rollout on shorter time steps until convergence is achieved. Then, we increase the training step and repeat the aforementioned process. Training on shorter time steps provides a good initialization for training on longer time steps, thereby improving computational efficiency. The balance of model effectiveness and computational efficiency brought about by VRAT can be intuitively seen through a simple case depicted in Fig. 7.

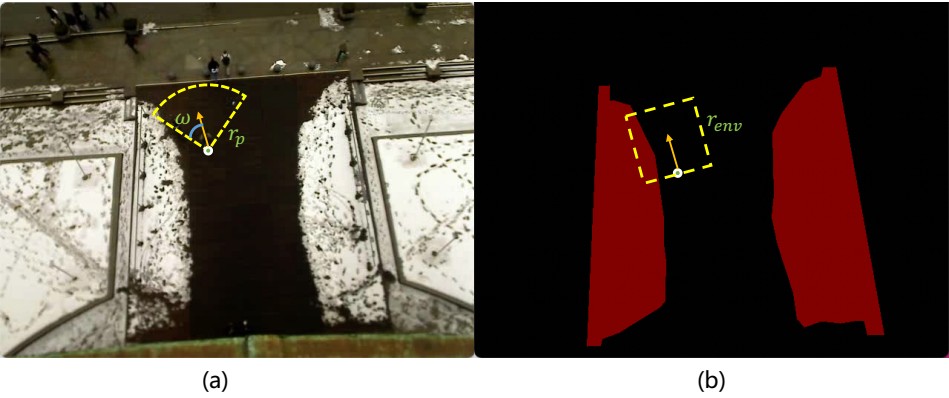

(a)        (b)

Figure 4: The shape of the pedestrian's field of view and its corresponding control parameters. (a) The field of view of a person for pedestrians and vehicles is a sector within a circle (centered at this person with radius $r_p$) spanned by an angle $\omega$ from the current velocity vector (orange arrow). (b) The environment is segmented into walkable (black) and unwalkable (red) areas. The view field is a square with dimension $r_{env}$ based on the current velocity vector (orange arrow)

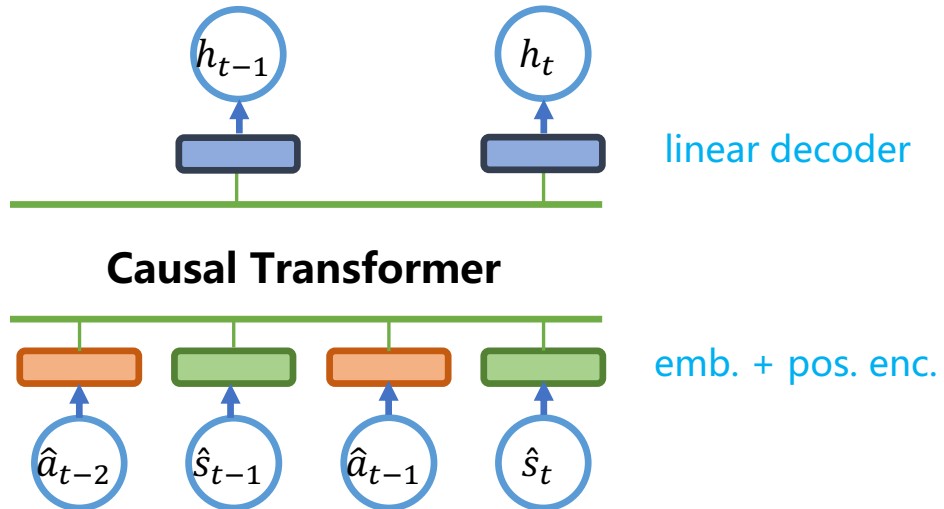

Figure 5: The input and output of the Transformer model in our framework. The input of the model consists of the states and corresponding actions of pedestrians in the past $n$ steps. The output of the model is a latent variable $h_t$ that includes the current moment information of pedestrians as well as their historical information. $h_t$ is used as input for the FIM and RA modules.

### A.3 LOSS FUNCTION

We propose a loss function to correct the model shift in VRTA. Here, we will further elaborate on the derivation process of our formula. As stated in the PROBLEM FORMULATION, the update of crowd is formulated as follows:

$$p_{t+1} = p_t + v_{t+1}\Delta t, v_{t+1} = v_t + a_{t+1}\Delta t \tag{9}$$

Therefore, the update mechanism for position can be expressed as follows:

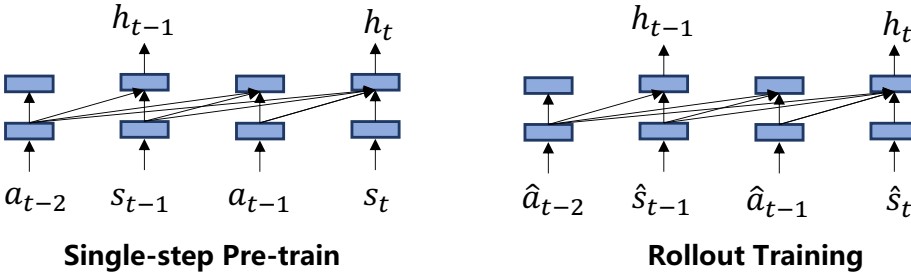

Figure 6: The differences between Single-step Pre-train and Rollout Training. In single-step Pre-train, only ground truth values are used as input. In contrast, multi-step training requires simulating a certain number of time steps and incorporating the information that is influenced by the model's prediction errors.

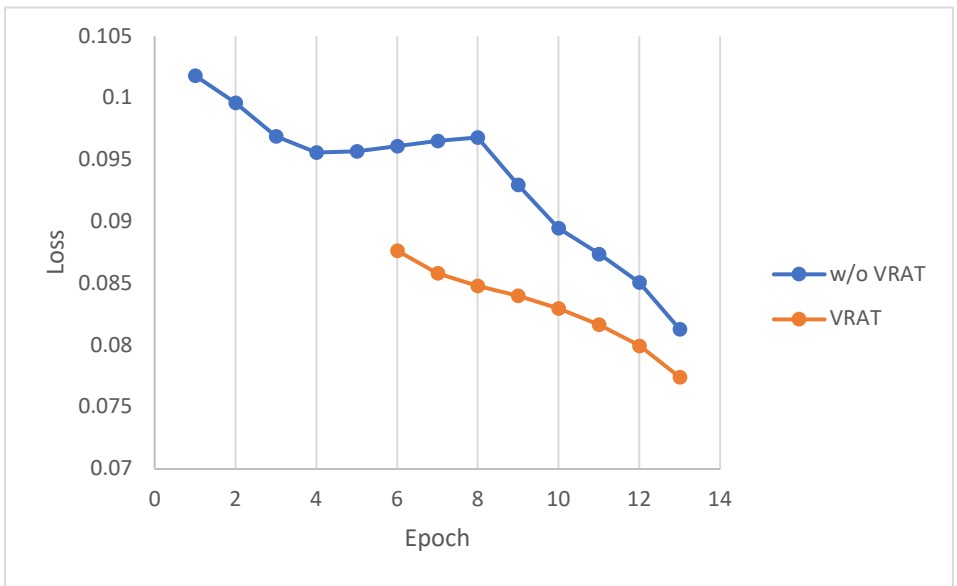

Figure 7: The role of VRAT in saving training time. In experiments without VRAT, the model is trained directly with rollouts at time steps of 20. However, in experiments with VRAT, the model is first trained with rollouts at time steps of 10 for a duration equivalent to 5 epochs at 20, before proceeding to perform rollouts at time steps of 20. It can be observed that performing rollouts at shorter time steps provides better model parameter initialization for rollouts at longer time steps, thus further balancing the effectiveness of the model with computational efficiency.

$$
\begin{aligned}
p_1 &= p_0 + v_1\Delta t \\
&= p_0 + (v_0 + a_1\Delta t)\Delta t \\
&= p_0 + v_0\Delta t + a_1\Delta t_2 \\
p_2 &= p_1 + v_2\Delta t \\
&= p_1 + (v_1 + a_2\Delta t)\Delta t \\
&= p_0 + (v_0 + a_1\Delta t)\Delta t + (v_0 + a_1\Delta t + a_2\Delta t)\Delta t \\
&= p_0 + 2v_0\Delta t^2 + 2a_1\Delta t^2 + a_2\Delta t^2 \\
p_3 &= p_0 + 3v_0\Delta t^2 + 3a_1\Delta t^2 + 2a_2\Delta t^2 + a_3\Delta t^2 \\
&\dots \\
p_{t+1} &= p_0 + (t+1)v_0\Delta t^2 + (t+1)a_1\Delta t^2 + ta_2\Delta t^2 + \dots + a_{t+1}\Delta t^2
\end{aligned}
\tag{10}
$$

After t simulation steps, the position of the pedestrian controlled by the model can be represented as follows:

$$\hat{p}_{t+1} = \hat{p}_0 + (t+1)v_0\Delta t^2 + (t+1)\hat{a}_1\Delta t^2 + t\hat{a}_2\Delta t^2 + ... + \hat{a}_{t+1}\Delta t^2 \tag{11}$$

The loss is calculated by the difference between $p_{t+1}$. and $\hat{p}_{t+1}$. Therefore, it is not only necessary to consider the differences between $a_{t+1}$ and $\hat{a}_{t+1}$, but also the losses incurred by historical behaviors, which can be represented as follows:

$$
\begin{aligned}
\mathcal{L} &= ||\hat{p}_{t+1} - p_{t+1}||^2 \\
&= \Delta t^4 ||(\hat{a}_{t+1} - a_{t+1}) + 2(\hat{a}_t - a_t) + ... + (t+1)(\hat{a}_1 - a_1)||^2 \\
&= \Delta t^4 ||\sum_{i=1}^{t} i \cdot (a_{t+1-i} - \hat{a}_{t+1-i})||^2
\end{aligned}
\tag{12}
$$

## A.4 BASELINES

As all existing learning-based and hybrid models need an observed sequence as input, for a fair comparison, we permit all models to observe each pedestrian for 8 steps

**Rule-based Models**: For Social Force Model (SFM), we let it observe each pedestrian for 8 steps and calculate the mean speed of each pedestrian as their desired walking speeds respectively, which makes it perform better than the classical SFM, in which all pedestrians have the same desired speed determined manually.

**Learning-based Models**: For all three data-driven models, We use their official implementations, convert our data to their accepted data format, and train the model with the observation length of 8 time steps and prediction length of 12 time steps. We then perform grid searches on the learning rate.

**Physics-infused Models**: For all three hybrid models, We use their official implementations, convert our data to their accepted data format. For NSP, we train it with the observation length of 8 time steps and prediction length of 12 time steps. We perform grid searches on the learning rate for NSP. We perform grid searches on the learning rate, rollout steps and regularization coefficient for PCS.

## A.5 SUPPLEMENTARY EXPERIMENT

We follow the experimental setup in Zhang et al. (2022). The experimental dataset are two public real-world datasets, including the GC dataset and the UCY dataset. We compare our framework with five state-of-the-art models from three categories, including physics-based models, learning-based models and physics-infused. Physics-based models include the Social Force Model (SFM) Helbing & Molnar (1995) and the Machine-Learning-Aided Physical Model (MLAPM) Zhang et al. (2022). Learning-based methods include Social-LSTM Alahi et al. (2016), Social-GAN Gupta et al. (2018), and Social-STGCNN Mohamed et al. (2020). Physcis-infused method is Physical-informed Crowd Simulator (PCS) Zhang et al. (2022)

For the GC dataset, we use three minutes for training, one for validation, and one for testing. For the UCY dataset, we use 108 seconds for training, 54 seconds for validation, and 54 seconds for testing. We use four widely adopted metrics, including the Mean Absolute Error (MAE), characterizing microscopic simulation accuracy, Optimal Transport divergence (OT) and Maximum Mean Discrepancy (MMD), measuring the differences between the generated simulation and the ground truth, and Collision (Col), characterizing the simulation's fidelity in terms of the collision avoidance behaviors. The performance is illustrated in Table A.5, our model outperforms the state-of-the-art (SOTA) in most metrics.

To examine whether our proposed method can generalize well beyond its training distributions, we test the performance of all methods trained on the GC Dataset on the UCY dataset and a new time period of the GC dataset, which we referred to as GC2 Dataset. Both datasets have a data distribution fundamentally different from the original GC dataset, and the degree of the differences is different. We show the results in Table A.5. Our models outperform all the baselines on both datasets, which demonstrates their strong generalizability.

| Groups | Models | GC | | | | UCY | | | |
|---|---|---|---|---|---|---|---|---|---|
| | | MAE | OT | MMD | Col | MAE | OT | MMD | Col |
| Rule -based | SFM | 1.259 | 2.114 | 0.015 | 622 | 2.539 | 6.571 | 0.129 | 434 |
| | MLAPM | 1.136 | _1.740_ | _0.012_ | **398** | 2.406 | 6.383 | 0.125 | **204** |
| Learning -based | Social-STGCNN | 7.669 | 20.31 | 0.613 | >9999 | 8.304 | 23.31 | 0.698 | >9999 |
| | Social-GAN | 7.513 | 25.21 | 0.387 | >9999 | 8.698 | 54.54 | 0.557 | >9999 |
| | Social-LSTM | 6.922 | 11.34 | 0.345 | >9999 | 7.291 | 16.41 | 0.476 | >9999 |
| Physics -infused | PCS | _1.097_ | 1.774 | 0.015 | 558 | _2.330_ | 6.250 | 0.109 | 264 |
| | Ours | **0.747** | **1.208** | **0.008** | _436_ | **1.327** | **3.800** | **0.047** | _226_ |

Table 6: The performance evaluation results on the GC and UCY datasets.

| Groups | Models | GC2 | | | | UCY | | | |
|---|---|---|---|---|---|---|---|---|---|
| | | MAE | OT | MMD | Col | MAE | OT | MMD | Col |
| Rule -based | SFM | 1.545 | 2.998 | 0.033 | 338 | 2.435 | 6.535 | 0.133 | 358 |
| | MLAPM | _1.394_ | _2.765_ | _0.029_ | **218** | 2.366 | 6.268 | 0.123 | **208** |
| Learning -based | Social-STGCNN | 7.691 | 24.61 | 0.509 | >9999 | 7.763 | 19.06 | 0.5719 | >9999 |
| | Social-GAN | 8.948 | 44.18 | 0.581 | >9999 | 6.884 | 32.25 | 0.512 | >9999 |
| | Social-LSTM | 6.889 | 14.63 | 0.387 | >9999 | 7.744 | 19.04 | 0.569 | >9999 |
| Physics -infused | PCS | 1.500 | 2.910 | 0.036 | 330 | _2.327_ | _6.165_ | 0.107 | 296 |
| | Ours | **0.745** | **2.108** | **0.016** | _230_ | **1.499** | **4.211** | **0.072** | _282_ |

Table 7: The generalization performance of the models trained on GC dataset.

In the aforementioned experiment, our approach consistently performed slightly lower than MLAPM in the collision metric. One major reason for this phenomenon is the explicit repulsion between pedestrians in rule-based methods, which significantly reduces the probability of collisions among pedestrians. In our model, there may also exist an attraction between pedestrians, similar to walking in groups, which makes pedestrians more likely to move in clusters.

## A.6 EVALUATION METRICS

Mean Absolute Error (MAE): It can be expressed as follows:

$$MAE = \frac{1}{N} \sum_{i=1}^{N} ||\hat{p}_i - p_i||^2 \tag{13}$$

where $N$ denotes the total number of predicted instances, $\hat{p}_i$ denotes the prediction of the pedestrian position and $p_i$ denotes the real position, and the $||\cdot||$ is the $l_2$ norm of a given vector.

Dynamic Time Warping (DTW): It is a widely used approach for comparing the similarity between two time series. The calculation method involves aligning the two series at the beginning and end, and then computing the cumulative pairwise distance between the elements of the two sequences as follows:

$$P = p_1, p_2, ..., p_n, \hat{P} = \hat{p}_1, \hat{p}_2, ..., \hat{p}_n \tag{14}$$

$$D(i, j) = d(\hat{p}_i, p_j) + min(D(i-1, j), D(i, j-1), D(i-1, j-1)) \tag{15}$$

where $d(\hat{p}_i, p_j)$ represents the measured distance between the i-th point of the predicted trajectory and the j-th element of real trajectory. $D(n, n)$ represents the optimal matching distance between the predicted trajectory and the real trajectory.

Optimal Transport (OT): It measures the distance between two distributions as the minimum cost to transport from distribution $P$ to distribution $Q$. Specifically, it has the form as

$$OT(P||Q) = \inf_{\pi} \int_{X \times Y} \pi(x,y)c(x,y)dxdy,$$

$$s.t. \int_{Y} \pi(x,y)dy = P(x), \int_{X} \pi(x,y)dx = Q(y), \tag{16}$$

where $\pi(x,y)$ can be approximated from $P(x)$ and $Q(y)$ by Sinkhorn Algorithm. In this work, We use 2D Wasserstein distance, i.e. $c(x,y) = ||x - y||_2^2$, and $X \equiv Y$ is the simulation duration, $P, Q$: $\mathbb{R} \to \mathbb{R}^2$ is the prediction trajectory and the true trajectory.

Maximum Mean Discrepancy (MMD): It takes the maximum difference between two distributions' moments in any order as their distance. It is mainly used to measure the distance between the distributions of two different but related random variables. Specifically, it has the form as

$$MMD[F, p, q] := \sup_{f \in F} (E_p[f(x)] - E_q[f(y)]) \tag{17}$$

where $p, q$ correspond to the predicted and the true position, respectively.

Collision: We count two pedestrians with a distance less than 0.5m as a collision and take the summation of collisions in all frames as Collision. But considering that pedestrians could walk with their friends, to whom they won't keep a large social distance, We take the pair of pedestrians that have a collision in more than 2 seconds as friends and do not count the collisions between them into Collision.

