# OpenReview forum: "Physics-infused Intention Network for Crowd Simulation"
_ICLR.cc/2024/Conference — Submitted to ICLR 2024_

### Official Review · Reviewer_j7Sd · 2023-10-21

**Soundness:** 3 good
**Presentation:** 4 excellent
**Contribution:** 3 good
**Rating:** 8
**Confidence:** 2

**Summary:**

The paper aims to improve crowd simulation methods, which are essential in various domains like traffic management, urban planning, and emergency management. Existing methods, rule-based and learning-based, have their limitations such as lack of authenticity and generalization, respectively. The authors introduce PINE, a framework that infuses physical biases and attention mechanisms into crowd simulation. PINE aims to enhance the authenticity and generalizability of crowd simulations by allowing pedestrians to adaptively adjust their behavior based on different influencing factors. PINE combines rule-based models with neural networks. It introduces modules like Feature Adaptive Extraction Module (FAEM) and Force Attention Module (FAM) to selectively extract and focus on relevant information and influences, improving the simulation's accuracy and physical consistency. A new Variable-step Rollout training algorithm is introduced to address cumulative errors during simulation, aiming to improve the model's performance further. Extensive experiments have been conducted, demonstrating that PINE outperforms state-of-the-art simulation methods in terms of accuracy, physical fidelity, and generalizability. Case studies are also visualized to help readers understand the advantages of this proposed method.

**Strengths:**

1. PINE introduces a novel framework that combines rule-based models with neural networks, infusing physical biases and attention mechanisms into crowd simulation. The way the physical model and the neural network are combined makes sense. The framework enhances the authenticity and generalizability of crowd simulations, allowing pedestrians to adaptively adjust their behavior based on different influencing factors. Feature Adaptive Extraction Module (FAEM) and Force Attention Module (FAM) have been introduced to selectively extract and focus on relevant information and influences to long trajectories. All the model design reflects the authors' original thinking and insights regarding the crowd simulation problem, which is a very important issue and approach. A new training algorithm is introduced to address cumulative errors during simulation, aiming to improve the model's performance further.

2. PINE has been extensively validated through experiments, demonstrating superior performance in terms of accuracy, physical fidelity, and generalizability compared to state-of-the-art simulation methods. I also enjoy the case study and visualization the authors give. The experiments are strong and promising. The generalization study and ablation study are comprehensive.

3. I can appreciate the way of incorporation of the physical model. To get the optimized force at the output end, the attention mechanism is used to obtain the coefficients for incorporating the forces to simulate the intent of the crowd. The residual action is then used as a “latent force” to help facilitate the limited capacity or missed information from the physical model. To me, it seems that the FAM part is quite interpretable. We will be able to understand each agent's action robustly. Moreover, I like that the problem is defined as a simulation, since the black-box model alone only tells us what to do. With simulation of pedestrians, for example, we can even cut out this particular part as a simulation environment (as long as it is stable and accurate) for the reinforcement learning of the vehicle agent to learn from.

**Weaknesses:**

1. While PINE aims to improve generalizability, it might still face challenges in adapting to various unforeseen scenarios or extreme conditions in real-world applications. These situations are really the main barriers and challenges in the field. It would be better to discuss more about such anomalies and whether the proposed model is robust enough.

2. Given the complexity and the number of components involved in the PINE framework, it might require significant computational resources for training and simulation. It would be better to have more information regarding the time complexity and time cost compared to previous methods and SFM methods.

**Questions:**

1. The paper mentions that uniform physical rules fail to effectively capture individual differences such as age, gender, and culture in pedestrian behavior. Since w/o residual action results in drastic performance degradation, relying too much on the physical model is indeed harmful. However, is residual action modeled by a single MLP enough? Do you need more advanced structure to help with that? Or, if MLP is way good enough, should we try a simpler RA model for better interpretability?

2. The so-called physical model of social force model is more like a self-defined “physical model” instead of the physical model for natural phenomenon such as the falling of apple or the precession of mercury. The authors are correct in a way that such a physical model has its limitations, so I agree that incorporating deep learning is necessary while the “physical model” provides stability and interpretability. I just wonder whether it would be better to have a more advanced “physical model” such as PDEs and symbolic regression to do the job, rather than the basic Newton's second law of motion (it might be too simple). There are some instances [1-4].

[1] Steven L Brunton, Joshua L Proctor, and J Nathan Kutz. 2016. Discovering governing equations from data by sparse identification of nonlinear dynamical systems. Proceedings of the national academy of sciences 113, 15 (2016), 3932–3937
[2] Udrescu, Silviu-Marian, and Max Tegmark. "AI Feynman: A physics-inspired method for symbolic regression." Science Advances 6.16 (2020): eaay2631.
[3] Michael Schmidt and Hod Lipson. 2009. Distilling free-form natural laws from experimental data. Science 324, 5923 (2009), 81–85
[4] Chen, Yuntian, et al. "Symbolic genetic algorithm for discovering open-form partial differential equations (SGA-PDE)." Physical Review Research 4.2 (2022): 023174.

I am not an expert in this specific area. My judgement is based on our understanding of similar tasks in Reinforcement Learning and Physics+AI.

---

> ### Author Response · Authors · 2023-11-17
>
> **We thank the reviewer for reading our paper carefully and giving the above positive. We hope that our answers to your questions can help you better understand our work. We also hope that our discussion can inspire future work in related fields.**
>
> > q1) The paper mentions that uniform physical rules fail to effectively capture individual differences such as age, gender, and culture in pedestrian behavior. Since w/o residual action results in drastic performance degradation, relying too much on the physical model is indeed harmful. However, is residual action modeled by a single MLP enough? Do you need more advanced structure to help with that? Or, if MLP is way good enough, should we try a simpler RA model for better interpretability?
>
> We greatly appreciate the reviewer for raising a thought-provoking question. As mentioned by the reviewer, pedestrian walking exhibits heterogeneity, and these heterogeneous attributes are encoded within the historical trajectories of pedestrian. While the final output of RA is a relatively simple MLP, its input consists of latent variables obtained from sequential model that process the historical trajectories of pedestrians. Through our experimental testing, we found that after sufficient training, RA is capable of effectively capturing the hidden heterogeneity within the historical information. Certainly, we do not deny that replacing MLP with other more sophisticated model structures may potentially yield better results. However, it is only an improvement within the existing framework. As desired by the reviewers, our ultimate goal is to capture heterogenous behavior with explicit physical significance. It is indeed regrettable that the information provided by existing public datasets is still limited, and we are unable to fully achieve the aforementioned objective. We believe our work can inspire future researchers to conduct more in-depth explorations and gather richer datasets in this direction.
>
> > q 2) The so-called physical model of social force model is more like a self-defined “physical model” instead of the physical model for natural phenomenon such as the falling of apple or the precession of mercury. The authors are correct in a way that such a physical model has its limitations, so I agree that incorporating deep learning is necessary while the “physical model” provides stability and interpretability. I just wonder whether it would be better to have a more advanced “physical model” such as PDEs and symbolic regression to do the job, rather than the basic Newton's second law of motion (it might be too simple). There are some instances [1-4].
>
> Thank you for the suggestions. Currently, SFM is widely recognized as an effective physical model in the field of pedestrian simulation. Despite its simple structure, the application of SFM in numerous commercial simulation software has demonstrated its effectiveness in pedestrian simulation. However, we also acknowledge that a more advanced “physical model” could be better to simulate pedestrian behavior. Indeed, pedestrian behavior involves highly complex mechanisms. SFM simplifies the characterization of these behavioral mechanisms to ensure model interpretability and generalizability. We believe that advancing the characterization of pedestrian behavior in physical models requires interdisciplinary collaboration. By further exploring the mechanisms of pedestrian behavior and incorporating this knowledge, we can develop more complex and interpretable physical models. We hope that our work can inspire progress in this area, and we are committed to furthering our efforts in this direction in the future.

---

> > ### Comment · Reviewer_j7Sd · 2023-11-23
> > **Official Reply**
> >
> > I have read the rebuttal by the authors. Thank you. I maintain my score, but lower my confidence to 2. I am not an expert in this specific area. My judgement is based on our understanding of similar tasks in Reinforcement Learning and Physics+AI.

---

### Official Review · Reviewer_nLKg · 2023-10-29

**Soundness:** 2 fair
**Presentation:** 2 fair
**Contribution:** 2 fair
**Rating:** 3
**Confidence:** 4

**Summary:**

The paper proposes a physical infused intention method to conduct crowd simulation. In the proposed work, the authors strengthen the importance of combining both advantages of rule-based and learning-based. Specifically, in the PINE network, a force attention module is designed to optimize the original force weight calculated by the Social Force Model. And a sequential module with a MLP are used to produce residual actions. They also introduced a variable step training process and designed a novel loss to reduce the cumulative error.

**Strengths:**

1. The paper is another approach intended to provide a seamless fusion of physic model and learning model in a crowd simulation scenario. The problem it intends to solve is significant and interesting.
2. The purpose of an additional term in the variable step rollout algorithm is innovative, intended to consider the bias of predicted actions and true actions, reflecting the model’s learning status to better guide the training process.

**Weaknesses:**

1. The paper lacks explicit explanation on important components, like the design of sequential models, how the attention model is updated, which is critical for understanding the rationale of the proposed work.
2. The paper’s writing is a bit vague in claiming other work’s weaknesses or shortcomings, authors are suggested to make more detailed analysis when claiming a point. E.g.,  In Section 4.5, When discussing the limitations of existing work, It is not clear why "initiating iteration at a fixed length" is computationally expensive. More details can be found in the Question part.
3. The experiment baselines do not sufficiently include the important baseline methods, e.g., the Social-GAN[1], Social-LSTM[2], Social-STGCNN[3], etc.

[1]:Gupta, Agrim, et al. "Social gan: Socially acceptable trajectories with generative adversarial networks." Proceedings of the IEEE conference on computer vision and pattern recognition. 2018.

[2]:Alahi, Alexandre, et al. "Social lstm: Human trajectory prediction in crowded spaces." Proceedings of the IEEE conference on computer vision and pattern recognition. 2016.

[3]:Mohamed, Abduallah, et al. "Social-stgcnn: A social spatio-temporal graph convolutional neural network for human trajectory prediction." Proceedings of the IEEE/CVF conference on computer vision and pattern recognition. 2020.

**Questions:**

1. Minor questions and suggestions:
- In the caption of Figure 1, the timeline seems to be the horizontal axis and vertical axis is the various methods.
- It is suggested to provide explicit descriptions, e.g., when authors describe the current exploration on a combination of rule based and learning based methods, author wrote:'They replace key components of the rule-based model with neural networks and train on real data.', where the ‘ key components' sound vague, what part is replaced? It is important to clarify when providing summary on existing methods.

2. In the paper, it is not clear on the implementation details of the sequential model, however, it is the crucial component to understand the rationality of later residual action generating. For example, In the equation (6), how does a simple MLP capture the intrinsic and stochastic factors in the behaviors? (i.e., whether the incorporated is the relative information, ect).

3. In Section 4.5, When discussing the limitation of existing work, It is not clear why "initiating iteration at a fixed length" is computationally expensive. It is suggested to provide profound analysis and discussion.
4. In equation (7), In the additional bias term, assume the i=100, the coefficient of the last item might be much greater than the actual gap between the actions, could the author explain more on the design here?
5. Based on the equation (5), How is the attention weight learned and updated? From what is written here, it seems to be a simple sigmoid function? Does the author imply any kind of standard attention machism? If not, the term is misleading.
6. In the experiment set ups, the baseline methods did not include some important methods like: Social-GAN, Social-LSTM, Social-STGCNN, etc. And the collision is not analyzed. Besides, one of the baseline methods: PCS does not seem to align with the original paper's performance on the same datasets GC and UCY. Could the author explain if is this caused by different settings or other factors?

---

> ### Author Response · Authors · 2023-11-17
>
> **We gratefully thanks for the precious time the reviewer spent making constructive remarks. We hope that our response will assist you in better assessing the value of our work.**
>
> > **Q1.1) In the caption of Figure 1, the timeline seems to be the horizontal axis and vertical axis is the various methods.**
>
> We appreciate the reviewer for pointing out the errors we made. The descriptions of the horizontal and vertical axes in the title of Figure 1 have been revised.
>
> > **Q1.2) It is suggested to provide explicit descriptions, e.g., when authors describe the current exploration on a combination of rule based and learning based methods, author wrote: 'They replace key components of the rule-based model with neural networks and train on real data.', where the ‘key components' sound vague, what part is replaced? It is important to clarify when providing summary on existing methods.**
>
> Thanks for pointing out the shortcomings. The more accurate description has been revised in the original text.
>
> They replace key components of the rule-based model, **such as partial or complete set of parameters of force components in SFM (Social Force Model)**, with neural networks and train on real data.
>
> > **Q 2) In the paper, it is not clear on the implementation details of the sequential model, however, it is the crucial component to understand the rationality of later residual action generating. For example, In the equation (6), how does a simple MLP capture the intrinsic and stochastic factors in the behaviors? (i.e., whether the incorporated is the relative information, ect).**
>
> We apologize for the unclear description of our framework. Indeed, the sequential model is an essential component in our framework, responsible for extracting pedestrian's historical trajectory information. The sequence model serves as a pluggable component within our framework. Our framework supports the integration of any type of sequence model, such as Transformer or LSTM. Specifically in our paper, we have adopted the Transformer as the sequential model in our framework. The historical states and actions of pedestrians are separately fed into the Transformer. The Transformer then processes this input and generates tokens that include the history information. A more detailed description and schematic diagram of the model structure is added to the appendix A.2.
>
> As for the residual action, we apologize for the inaccuracy in its description. In fact, residual action is used to describe the factors influencing pedestrian movement that are not covered by SFM (Social Force Model) structure, such as psychological factors. These latent factors can be mined through the pedestrians' historical trajectory information. In our framework, the sequence model plays a role in mining the information, while the MLP acts as the head, mapping the information extracted by the sequence model to the dimensions of actions. We have revised the inaccurately described portion in the original text, and we are grateful for the suggestions from the reviewer.
>
> > **Q 3) In Section 4.5, When discussing the limitation of existing work, It is not clear why "initiating iteration at a fixed length" is computationally expensive. It is suggested to provide profound analysis and discussion.**
>
> Similar suggestions were also raised by Reviewer 9dce, you can refer to the W2) for a more detailed explanation of the motivation behind VRAT. We apologize for the inaccuracy description in Section 4.5. Actually, what we want to express is achieving a better balance between computational efficiency and model performance.
>
> Before designing VRAT, we attempted to perform long-step rollout training after single-step pre-train. However, we found that this approach required a significant amount of time to train the model until convergence. Performing rollouts on shorter time steps proved to be an effective strategy in reducing training time and providing better model parameter initialization for rollouts on longer time steps. This approach improves the efficiency of rollouts on longer time steps.
>
> Therefore, our proposed VRAT approach achieves a better balance between model performance and training efficiency. The relevant description has been modified in the original paper and highlighted in blue. A more intuitive visualization of the difference in loss reduction is provided in Appendix A.2.

---

> ### Author Response · Authors · 2023-11-17
>
> > **Q 4) In equation (7), In the additional bias term, assume the i=100, the coefficient of the last item might be much greater than the actual gap between the actions, could the author explain more on the design here?**
>
> We appreciate the reviewer for bringing up a highly worthwhile question for discussion. When designing the additional bias, we have also carefully considered the situation where the additional bias occurs due to accumulated errors when the simulation steps are too high. In our experiment, we limit the maximum value of i to 20 to prevent the aforementioned issue.
>
> We impose a restriction on the maximum step length for two main reasons. Firstly, it is due to the training mechanism of the Rollout itself. When the training step length is too long, the accumulated errors can cause pedestrians to deviate significantly from the ground truth states in the dataset. In such cases, it is not reasonable to let the model learn the corresponding behaviors in the dataset. Because the true behavior of pedestrians in the current state is unknown. The second point is to consider the generalizability of the model. The model learns better from data that corresponds to an appropriate number of steps. Training on longer datasets can easily lead to overfitting the model to that specific dataset. In PCS [1], the authors also adopted a similar approach of using close training step lengths.
>
> [1] Zhang, Guozhen, et al. "Physics-infused machine learning for crowd simulation." Proceedings of the 28th ACM SIGKDD Conference on Knowledge Discovery and Data Mining. 2022.
>
> > **Q 5) Based on the equation (5), How is the attention weight learned and updated? From what is written here, it seems to be a simple sigmoid function? Does the author imply any kind of standard attention machism? If not, the term is misleading.**
>
> We apologize for any confusion caused by naming the module as "attention." In fact, the physical meaning of this module is closer to the "intention" mentioned in our title. The intention of pedestrians should indeed be more flexibly adjusted. For example, in the scenario mentioned in the paper where pedestrians avoid vehicles, the forces that direct pedestrians towards their goal and repel them from the vehicle should both be reduced to zero, while the forces that facilitate avoidance of other pedestrians should remain unchanged. Therefore, it should not be a standard attention module where the weights sum up to 1. To achieve the physical process as described above, we have employed a sigmoid-like gating structure. It’s sample but effect. We have made modifications to the nomenclature of the relevant modules in the paper in order to prevent any possible reader misunderstandings.
>
> > **Q 6) In the experiment set ups, the baseline methods did not include some important methods like: Social-GAN, Social-LSTM, Social-STGCNN, etc. And the collision is not analyzed. Besides, one of the baseline methods: PCS does not seem to align with the original paper's performance on the same datasets GC and UCY. Could the author explain if is this caused by different settings or other factors?**
>
> We apologize for the lack of clarity in our experimental setup statement. The differences observed in the performance of the PCS method in our experiment are related to the data set partitioning and training settings. In the GC and UCY scenarios, we selected complete trajectories within the entire scene instead of performing partial trajectory truncation as described in the original paper, because we consider that more complete and longer trajectories pose a greater challenge to the simulation algorithm. On the other hand, we only used the GC and UCY datasets to test the generalization of our model. The model was trained on DUT and ETH datasets, which represent completely different scenarios. The aforementioned factors resulted in the observed differences in the performance of the PCS method.
>
> Following the reviewer's suggestion, we have included the results with the same experimental settings as the original article in the appendix A. 5. Important methods including Social-GAN, Social-LSTM, Social-STGCNN and Collision metric are all included.

---

> ### Author Response · Authors · 2023-11-17
> **Supplementary Experiment**
>
> **The performance evaluation results on the GC and UCY datasets.**
> |               |       | GC    |       |      |       | UCY   |       |     |
> |---------------|-------|-------|-------|------|-------|-------|-------|-----|
> | **Models**        | **MAE**   | **OT**    | **MMD**   | **Col**  | **MAE**   | **OT**    | **MMD**   | **Col** |
> | SFM           | 1.259 | 2.114 | 0.015 | 622  | 2.539 | 6.571 | 0.129 | 434 |
> | MLAPM         | 1.136 | $ \underline{1.740}$ | $ \underline{0.012}$ | **398**  | 2.406 | 6.383 | 0.125 | **204** |
> | Social-STGCNN | 7.669 | 20.31| 0.613| >9999 | 8.304 | 23.31 |0.698 |>9999|
> | Social-GAN    | 7.513 |25.21 |0.387 | >9999| 8.698| 54.54 |0.557 |>9999 |
> | Social-LSTM   | 6.922 |11.34| 0.345| >9999| 7.291 |16.41| 0.476 |>9999 |
> | PCS           |$ \underline{1.097}$ |1.774 |0.015 | 558 |$\underline{2.330}$ |$\underline{6.250}$ |$\underline{0.109}$ |264 |
> | Ours          | **0.747** |**1.208** |**0.008** |$\underline{436}$ |**1.327** |**3.800** |**0.047** |$\underline{226}$|
> |               |       |       |       |      |       |       |       |     |
>
>
> **The generalization performance of the models trained on GC dataset.**
> |               |       | GC    |       |      |       | UCY   |       |     |
> |---------------|-------|-------|-------|------|-------|-------|-------|-----|
> | **Models**        | **MAE**   | **OT**    | **MMD**   | **Col**  | **MAE**   | **OT**    | **MMD**   | **Col** |
> | SFM           | 1.545 | 2.998 | 0.033 | 338 | 2.435 | 6.535 | 0.133 | 358 |
> | MLAPM         | $ \underline{1.394}$ | $ \underline{2.765}$ | $ \underline{0.029}$ | **218** | 2.366 | 6.268 | 0.123 | **208** |
> | Social-STGCNN | 7.691 |24.61 |0.509 | >9999 | 7.763 | 19.06 | 0.5719 | >9999|
> | Social-GAN    | 8.948 | 44.18 | 0.581 | >9999| 6.884| 32.25| 0.512| >9999 |
> | Social-LSTM   | 6.889 | 14.63 | 0.387 | >9999 | 7.744 | 19.04 | 0.569 | >9999 |
> | PCS           |1.500 | 2.910 | 0.036 | 330 | $ \underline{2.327}$ | $ \underline{6.165}$ | $ \underline{0.107}$ | 296|
> | Ours          | **0.745** |**2.108**| **0.016**|$ \underline{ 230}$| **1.499**| **4.211**| **0.072**| $ \underline{282}$|
> |               |       |       |       |      |       |       |       |     |

---

### Official Review · Reviewer_9dce · 2023-11-01

**Soundness:** 3 good
**Presentation:** 3 good
**Contribution:** 3 good
**Rating:** 5
**Confidence:** 3

**Summary:**

This paper proposes a crowd simulation model called Physics-infused Intention NEtwork (PINE) combining the merits of rule-based and learning-based methods to improve the authenticity and generalization performance of the simulation. The contributions of this research are as follows:
1) A proposal of PINE for crowd simulation
2) An introduction to the Variable-step Rollout Training algorithm
3) Conduction of extensive experiments

**Strengths:**

1. The authors evaluated their crowd simulation with multiple datasets to validate their model under various conditions.
2. they compared their framework with various models from different categories making their proposal reasonable.

**Weaknesses:**

1. More descriptions seem to be required for input data such as s_p and  s_e.
2. Variable-step Rollout Training Algorithm (VRTA) is one of the main proposals, but explanation is not sufficient to understand.

**Questions:**

1. What is the dimension of the s_p and  s_e considering the minimum dataset size?
2. Using VRTA, which mechanism is applied to incrementally increase the value of T for the purpose of saving computation?
3. For additional bias in VRTA, the latest value is multiplied by the biggest value by multiplying i to the subtraction term. What is the exact effect of multiplying i?

---

> ### Author Response · Authors · 2023-11-17
>
> **We gratefully appreciate for your valuable comments. We hope our answers to the 3 questions will address the concerns and clarify the contributions of the paper.**
> **If you have any further inquiries or suggestions, please do not hesitate to reach out to us. Detailed response are as below.**
> > **Q1) What is the dimension of the $s_p$ and $s_e$ considering the minimum dataset size?**
>
> Thank you for pointing out the deficiencies in our description of $s_p$ and $s_e$. This question bears resemblance to one weakness, namely the insufficient description for the variables $s_p$ and $s_e$. In our paper, we descriped $s_p$ and $s_e$ in **PROBLEM FORMULATION** section, presenting all internal factors and external factors that influence pedestrian walking, respectively. Specifically in our paper, $s_p$ represents the information regarding the pedestrian's own destination, $x^g$ in our **METHOD** section. The minimum dimension of $x^g$ is 6, including the relative position, relative velocity, and relative acceleration of the destination with respect to the pedestrian in a two-dimensional space. $s_e$ presents the surrounding pedestrians and obstacles around the pedestrian, $x^p$ and $x^o$ in our **METHOD** section. The calculation method for $x^p$ and $x^o$ is similar to $x^g$, and the minimal dimension of $s_e$ is 12.
>
> $s_p$ and $s_e$ are abstract representations of all factors influencing pedestrian walking behavior, and they can capture higher-dimensional information when the actual dataset has sufficient feature characteristics. Due to the limited feature quantity in publicly available datasets, we only included the aforementioned three types of information that directly affect pedestrian walking. The descriptive forms of s_p and s_e have also been adopted in similar works, such as in [1].
>
> > **W2) Variable-step Rollout Training Algorithm (VRTA) is one of the main proposals, but explanation is not sufficient to understand.**
>
> Thank you for pointing out the lack of clarity in our exposition on VART.
>
> In the single-step training process, the model only receives the ground-truth and outputs a single-step action to calculate the loss. However, during the simulation process, the model needs to iteratively output actions continuously. The previous action taken by the model will affect the next input of the model. This can potentially expose the model to a dataset that does not exist during training, leading to the occurrence of cumulative errors as the model continues to make incorrect actions.
>
> Rollout training, on the other hand, needs to run the model in the simulation environment for multiple steps before performing loss function computation and model parameter updates, brings the model closer to the real simulation scenario. However, this training method, similar to on-policy reinforcement learning, suffer from the issue of low sampling efficiency. The model needs to first simulate and sample a batch of data, calculate the error based on this batch of data, and update the model parameters accordingly. After the update, this batch of simulated data cannot be used for further updates. Therefore, the process of simulating and updating needs to be repeated until the model converges. This update process can consume a significant amount of time on sampling. To improve computational efficiency, it is common to combine single-step and multi-step. This means that the model parameters are first quickly initialized at a good starting point using single-step training method, and then rollout updates are used to further reduce the compounding error [1].
>
> There is a trade-off between training efficiency and model performance when it comes to rollout step. Specifically, the step size of rollout training is positively correlated with the model performance, as a longer step size brings the training closer to the real simulation scenario. However, in contrast, the longer the time step, the more time it takes to sample data in a single instance. In order to balance the training time and model performance, we introduced VRTA. In contrast to performing rollout directly on long time steps, we initially conduct rollout on shorter time steps. Once the training process approaches convergence, we gradually increase the time step length. Rollout training on shorter time steps offers higher sampling efficiency and promotes better model convergence. It also provides a well-initialized set of model parameters for rollout on longer time steps. As a result, VRAT achieves a better balance between computational efficiency and model performance.
>
> Thanks for the reviewer's suggestions. We have improved the description of VRAT in the corresponding section of the paper and provided additional explanations in the appendix A.2. All revisions are highlighted in blue font.
>
> [1] Zhang, Guozhen, et al. "Physics-infused machine learning for crowd simulation." Proceedings of the 28th ACM SIGKDD Conference on Knowledge Discovery and Data Mining. 2022.

---

> ### Author Response · Authors · 2023-11-17
>
> > **Q2) Using VRTA, which mechanism is applied to incrementally increase the value of T for the purpose of saving computation?**
>
> In our experimental setup, we adopted a linearly increasing T method. Once the model converges on shorter time steps, we linearly increase the size of T and perform rollout on longer time steps. We repeat this process until we reach our maximum T. The mechanism for saving computation has been explained in detail in W2. A more intuitive visualization of the difference in loss reduction is provided in Appendix A.2.
>
> > **Q3) For additional bias in VRTA, the latest value is multiplied by the biggest value by multiplying i to the subtraction term. What is the exact effect of multiplying i?**
>
> Thank you for pointing out the deficiencies in our derivation of the loss function. Here, we will further elaborate on the derivation process of our formula.
>
> As stated in the PROBLEM FORMULATION, the update of crowd is formulated as follows:
>
> \begin{equation}
> p_{t+1} = p_t +v_{t+1}\Delta t, v_{t+1}=v_t+a_{t+1}\Delta t
> \end{equation}
>
> Therefore, the update mechanism for position can be expressed as follows:
>
> \begin{equation}
> p_1 =p_0+v_1\Delta t =p_0+(v_0+a_1\Delta t)\Delta t =p_0+v_0\Delta t+a_1\Delta t_2
> \end{equation}
>
> \begin{equation}
> p_2 =p_1+v_2\Delta t =p_1+(v_1+a_2\Delta t)\Delta t=p_0+ (v_0+a_1\Delta t)\Delta t+(v_0+a_1\Delta t+a_2\Delta t)\Delta t=p_0+2v_0\Delta t^2+2a_1\Delta t^2+a_2\Delta t^2
> \end{equation}
>
> \begin{equation}
> p_3 =p_0+3v_0\Delta t^2+3a_1\Delta t^2+2a_2\Delta t^2+a_3\Delta t^2
> \end{equation}
>
> ...
>
> \begin{equation}
> p_t =p_0+tv_0\Delta t^2+ta_1\Delta t^2+(t-1)a_2\Delta t^2+...+a_t\Delta t^2
> \end{equation}
>
> After t simulation steps, the position of the pedestrian controlled by the model can be represented as follows:
>
> \begin{equation}
> \hat{p}_t=\hat{p}_0+tv_0\Delta t^2+t\hat{a}_1\Delta t^2+(t-1)\hat{a}_2\Delta t^2+...+\hat{a}_t\Delta t^2
> \end{equation}
>
> The loss is calculated by the difference between $p_t$ and $\hat{p}_t$.
>
> Therefore, it is not only necessary to consider the differences between $a_t$ and $\hat{a}_t$, but also the losses incurred by historical behaviors, which can be represented as follows:
>
> $$L =||\hat{p}_t-p_t||^2$$
>
> $$=\Delta t^4||(\hat{a}_t-a_t)+...+t(\hat{a}_1-a_1)||^2$$
>
> $$=||\sum_{i=1}^ti(a_{t+1-i}-\hat{a}_{t+1-i})||^2 $$
>
> The effect of multiplying by i can be seen as giving more weight to actions taken earlier, implying that early actions have a greater impact on the subsequent simulation performance of the model.
>
> Thank you very much for pointing out the deficiencies in our explanation of the formula. We have included the above derivation process in the appendix A.3.

---

### Author Response · Authors · 2023-11-22

Dear reviewers:

We extend our deepest appreciation for the perceptive and constructive critiques provided by the esteemed panel. It is with great satisfaction that we note your recognition of the significance and intrigue of our research topic [nLKg, j7Sd], the novelty and efficacy of our methodology [9dce, nLKg, j7Sd], and the effectiveness of our evaluation experiments [9dce].

In response to the concerns and issues raised, we have diligently endeavoured to fortify and enhance the accessibility of our research. These enhancements have been thoroughly integrated into the revised manuscript.

Since we are in the last two days of the discussion phase, we are eagerly looking forward to your responses. Please let us know if there are any additional clarifications or experiments that we can offer. We would love to discuss more if any concern still remains.

Best,

Authors

---

### Meta-Review · Area_Chair_HtQf · 2023-12-07

**Metareview:**

This paper demonstrates that the integration of rule-based and learning-based approaches in crowd simulation improves simulation results. It was pointed out that while the paper approaches an interesting problem of applied importance through a variety of technical efforts, the key ideas and details are not sufficiently convincingly explained. It is acknowledged that the paper itself has improved as a result of the communications with the reviewers. However, given the competitive nature of the conference,  the paper still has room for improvement, relatively speaking.

**Justification For Why Not Higher Score:**

This paper does not adequately present the ideas and effectiveness when considering the competitive nature of ICLR and the audience.

**Justification For Why Not Lower Score:**

NA

---

### Decision · Program_Chairs · 2024-01-16

Reject